# Metabolite Transporters as Regulators of Immunity

**DOI:** 10.3390/metabo10100418

**Published:** 2020-10-19

**Authors:** Hauke J. Weiss, Stefano Angiari

**Affiliations:** School of Biochemistry and Immunology, Trinity Biomedical Sciences Institute, Trinity College Dublin, Dublin 2, Ireland; angiaris@tcd.ie

**Keywords:** cell metabolism, immunity, metabolite transporter, solute carrier, T cells, macrophages

## Abstract

In the past decade, the rise of immunometabolism has fundamentally reshaped the face of immunology. As the functions and properties of many (immuno)metabolites have now been well described, their exchange among cells and their environment have only recently sparked the interest of immunologists. While many metabolites bind specific receptors to induce signaling cascades, some are actively exchanged between cells to communicate, or induce metabolic reprograming. In this review, we give an overview about how active metabolite transport impacts immune cell function and shapes immunological responses. We present some examples of how specific transporters feed into metabolic pathways and initiate intracellular signaling events in immune cells. In particular, we focus on the role of metabolite transporters in the activation and effector functions of T cells and macrophages, as prototype adaptive and innate immune cell populations.

## 1. Metabolites and Their Transporters: Key Players in Immunity

Active metabolite transport is key for the survival of every cell. In immune cells, availability of extracellular metabolites also shapes immune responses and cellular interactions. Metabolite transporters indeed largely dictate immune cell activity by providing, or restricting, access to nutrients, and thereby determining the cellular metabolic profile [1]. For example, while glucose and amino acids (AAs) are largely imported as part of pro-inflammatory responses, fatty acid (FA) uptake has been found to drive pro-resolving, anti-inflammatory phenotypes [2]. Importantly, expression of the respective transporters is tightly linked with metabolic reprogramming in immune cells, as evident by direct involvement in the activation of metabolic master regulators, such as the mammalian target of rapamycin complex (mTORC) [3]. Furthermore, metabolites are actively released into or scavenged from the extracellular space by some cells to drive metabolic changes in their environment. A prime example is the Warburg effect, where cancer cells switch away from aerobic respiration towards fermentation of glucose to quickly generate biomass [4,5]. At the same time, release of “leftover” metabolites such as lactate hugely affects the metabolism and fate of surrounding immune cells [6].

As cellular metabolism itself, metabolite transporters have been well studied for decades, with their main function mostly being attributed to providing energy sources and building bricks for cells. Being polar molecules, many metabolites require active transport across the cell membrane, as they are unable to readily diffuse into the cell [7]. Hence, regulation and expression of dedicated transporters is critical to meet cell metabolic needs and initiate alternative pathways to adapt to changes in the cellular environment. As rate-limiting providers of nutrients, metabolite transporters have also gained interest as putative drug targets in recent years, with many potential drugs with immunomodulatory function being currently in development.

In mammals, metabolites are mostly transported across the plasma membrane by members of the solute carrier (SLC) family of membrane transport proteins, many of which are expressed in immune cells. More than 400 individual SLCs have been discovered, grouped into 52 subfamilies [8]. SLCs transport substrates through one of three modes: (i) cotransporters need to translocate their substrate together with an ion, which travels along a concentration gradient, thus providing the energy required; (ii) exchangers use the same mechanism, but here, the ion and substrate travel in opposite directions; and (iii) facilitated transporters translocate a single substrate passively along a concentration gradient. However, as there are still many orphan transporters, i.e., transporters of unknown function and with unknown substrates, additional modes of transport might exist in the SLC family [9,10]. Other than metabolites, SLCs also transport metals, vitamins and neurotransmitters, as well as inorganic ions, confirming their crucial role in maintaining cell survival and regulating cell functionality [8]. 

In immune cells, expression of SLCs is tightly regulated. Many different SLCs have been shown to control sugar, lipid and amino acid flux in and out of immune cells, thus determining cell fate and function. In this review, we summarize the importance of such transporters in the activation, function and polarization of adaptive and innate immune cells, with a particular focus on T cells and macrophages.

## 2. Expression and Importance of Metabolite Transporters in T Cells

During differentiation and activation, T cells undergo distinct metabolic changes that cater to their respective needs. While naïve T cells mostly rely on the tricarboxylic acid (TCA) cycle and oxidative phosphorylation (OXPHOS), activated T cells switch to glycolysis, glutaminolysis and FA synthesis to accommodate an increased demand for energy [11]. There are also metabolic differences in different T cell subsets. As an example, regulatory T cells (Tregs) utilize FA oxidation (FAO) to generate energy when proliferating, while conventional CD4^+^ T cells mostly use glucose as fuel [12]. Abundance of these resources is thus crucial for T cell fate decision making, and expression of the respective transporter molecules is closely linked to T cell activity (Figure 1). 

### 2.1. Glucose Transport

Surface expression of the glucose transporter GLUT1 (also known as solute carrier family 2, facilitated glucose transporter member 1, SLC2A1) is induced upon T cell receptor (TCR) engagement [13], as well as signaling through CD28 [14]. Upregulation of GLUT1 is a crucial step in T cell activation, importing glucose to be metabolized through aerobic glycolysis, and restriction of glucose uptake has been found to largely impair T cell activation [14]. In addition to activated T cells, GLUT1 surface expression is also increased in naïve T cells upon interleukin 7 (IL-7) stimulation, and in memory CD4^+^ T cells after insulin treatment [15]. It is suggested that GLUT1 expression is crucial for development and maintenance of both CD4^+^ and CD8^+^ T cell memory [15,16]. Furthermore, CD4^+^ T cells require GLUT1 expression to differentiate into T helper type 1 (T_H_1), T_H_2 and T_H_17 cells [17], while overexpression of GLUT1 does not affect their development [14]. Other glucose transporters have been found to be expressed in the context of T cell activation. In addition to GLUT1, the glucose transporters GLUT3, GLUT4 and GLUT6 (SLC2A3, SLC2A4 and SLC2A6) have also been found to be upregulated in activated CD4^+^ T cells to varying degrees [18,19]. As GLUT1 is the most prominent and well-studied glucose transporter, it is believed that the other GLUT variants have mostly supporting or compensatory functions in most T cells [17]. As an exception, renal cell carcinoma (RCC) CD8^+^ tumor-infiltrating lymphocytes (TILs) specifically upregulate GLUT3 in response to CD28 co-stimulation [20].

Of note, besides providing nutrients, glucose import and metabolism have other distinct impacts on T cell functionality. The glycolytic enzyme glyceraldehyde 3-phosphate dehydrogenase (GAPDH) has been found to have a moonlighting function as a negative regulator of interferon gamma (IFN-γ) production in the absence of glycolysis [21]. Moreover, some products of glycolysis are involved in epigenetic modifications: pyruvate is processed into acetyl-coenzyme A (acetyl-CoA), which is not only oxidized for adenosine triphosphate (ATP) production in the mitochondria, but also involved in histone acetylation [22]. Furthermore, products of glycolysis can glycosylate proteins, modifying their function and localization and ultimately impacting T cell differentiation and activation [23]. Overall, regulation of glucose import and processing by GLUTs represents an essential step for T cell activation and engagement of effector functions.

### 2.2. Lactate Transport

Lactate often builds up in hypoxic conditions and at sites of inflammation [24]. A by-product of glycolysis, lactate, is generated from pyruvate by lactate dehydrogenase (LDH) as part of aerobic glycolysis [25], and can be further metabolized through the TCA cycle for energy generation [26]. Transport of lactate is facilitated by several transporters: proton-linked monocarboxylic acid transporters (MCTs) 1-4 are all bidirectional, however MCT1 (SLC16A1) is mostly relevant for lactate uptake, whereas MCT4 (SLC16A3) mostly exports excess lactate into the extracellular space [27]. Additionally, two other members of the SLC5 family, SLC5A12 and SLC5A8, can transport lactate [28]. In T cells, MCT1 expression is upregulated upon TCR activation, and its blockage results in impaired cytotoxic T cell (CTL) function in the tumor microenvironment [29]. Excess lactate import into T cells has also been shown to restrict proliferation and cytokine production of CTLs. This is possibly due to their reliance on glycolysis [30], as lactate import goes along with consumption of reduced nicotinamide adenine dinucleotide (NADH) and inhibition of glycolysis [31]. On the other hand, Tregs are able to change their metabolism to function in a lactate-high environment, and are therefore unaffected [32]. Recently, it has been found that lactate uptake via SLC5A12 results in pyruvate kinase M2/signal transducer and activator of transcription 3 (PKM2/STAT3)-mediated production of IL-17 in CD4^+^ T cells in rheumatoid arthritis (RA) [33]. Imported lactate fuels FA synthesis in these cells, which is indeed increased while they are retained at the site of inflammation, and blockage of SLC5A12 ameliorated the severity of the disease [33]. Similar observations were made previously in the tumor microenvironment, where lactate is imported into CD4^+^ T cells by SLC5A12, decreasing CD4^+^ T cell motility while inducing IL-17 production [34]. These works suggest a complex role for extracellular lactate uptake in controlling T cell fate and function, which may be particularly relevant at the site of inflammation [35]. 

### 2.3. Fatty Acid Transport

FAO is halted in activated T cells, and FA synthesis is initiated to provide growing and dividing cells with lipids [11]. Additionally, FAs are transported into activated T cells either by active diffusion [36], or through dedicated transporters. One is the FA translocase CD36 [37] that is also involved in uptake of other molecules, such as lipoproteins [38], oxidized phospholipids [39] and collagen [40]. CD36 is notably upregulated in Tregs, especially in those found in the tumor microenvironment [41]. A more specific uptake of FAs in T cells is through members of the FA transport protein (FATP) family. These transporters take up very long-chain FAs (VLCFAs) and simultaneously esterificate them to trap them inside the cell [42]. The roles of FATP1 and CD36 in T cell function are yet to be clearly determined, as it has proven difficult to separate imported FAs from those produced in metabolic pathways [1]. Shorter FAs can also be imported by a number of G protein-coupled receptors (GPCRs): GPCR40, 41, 43, 84 and 120 all have different affinities for certain FA lengths [43], however only GPCR43 and 84 are expressed in T cells. GPCR43 is crucial for maintaining intestinal homeostasis by inducing IL-10 production in intestinal T cells [44], while GPCR84 regulates IL-4 production in response to CD3 crosslinking, importing medium-chain FAs [45,46]. 

Acetate is another short-chain fatty acid that is imported into T cells through members of the MCT family or aquaporins [47,48,49]. In CD8^+^ T cells, acetate influx is required for optimal memory function: upon entry, acetate is converted to acetyl-CoA, subsequently acetylating and activating GAPDH, and thereby boosting memory CD8^+^ T cell responses [50]. Moreover, acetate-derived acetyl-CoA has been found to promote IFN-γ production in CD4^+^ T cells [51]. More recently, Qiu et al. found acetate imported through MCTs to restore effector function in glucose-restricted CD8^+^ T cells in cancer. Acetate effectively promoted histone acetylation via acetyl-CoA, resulting in increased transcription of IFN-γ and increased tumor clearance [52]. Interestingly however, high acetate levels at sites of infection have also been shown to augment glutaminolysis while simultaneously reducing calcium influx and effector function in CD8^+^ T cells, resulting in decreased immunopathology, yet improved pathogen clearance [53]. FA import thus regulates T cell functionality, but further investigation is needed to clarify the exact role of different FAs in the control of T cell activity.

### 2.4. Amino Acid Transport

Amino acids (AAs) are another highly valuable energy source for activated T cells, as they are a valuable resource for protein production. While the relevance of AA transport in developing and naïve T cells is postulated to be relatively low [54], T cell activation and differentiation into effector subsets largely depend on the abundance of extracellular AAs. Dedicated AA transporters like the large neutral amino acid transporter 1 (LAT1, also known as SLC7A5) and the sodium-coupled neutral amino acid transporters (SNATs) SNAT-1/SLC38A1 and SNAT-2/SLC38A2, and Alanine, Serine, Cysteine Transporter 2 (ASCT2/SLC1A5) are increased in expression in activated T cells [55,56,57]. Although leucine is the primary substrate for LAT1, this transporter is also able to translocate other AAs with a lower affinity. Forming a heterodimer with LAT-2/SLC7A8 in a complex with CD98 [58], LAT1 expression is required for a full CD8^+^ T cell response [59]. LAT1 is upregulated upon TCR stimulation through the extracellular signal-regulated kinase 1/2-mitogen-activated protein kinase (ERK/MAPK) pathway and activation of the nuclear factor of activated T cells (NFAT) [59], while c-Myc also regulates its expression [60]. Leucin is a critical factor in mTORC1 activation, and deletion of LAT1 results in severely impaired T cell activation [59] and cytokine production [61]. 

Despite its name, ASCT2 is mainly known as a glutamine transporter [62]. Similarly to LAT1, ASCT2-mediated glutamine uptake is required for TCR- and CD28-induced activation of mTORC1 in CD4^+^, but not CD8^+^ activated T cells [57]. Besides being metabolized for ATP generation in the TCA cycle, glutamine can be used as a substrate for production of other AAs, as well as polyamines and glutathione. The latter is an antioxidant and metabolic regulator, priming T cells for activation and enrolling them in a glycolytic metabolic program [63]. Furthermore, glutamine directly fuels LAT1 activity, because, as an antiporter, LAT1 requires glutamine as an efflux substrate [64]. ASCT2 is an antiporter itself, exporting the small AAs serine, asparagine or threonine [62]. However, how this ASCT2-mediated efflux of important immunomodulatory AAs such as serine [65] affects T cell biology is not known. Extracellular serine has also been found to enter T cells and modulate proliferation [65], but the transporter relevant for serine uptake remains to be identified. AA uptake through both LAT1 and ASCT2 has also been shown to be crucial for differentiation of CD4^+^ T cells into T_H_1 and T_H_17 cells, while T_H_2 cells and Tregs develop independently of these transporters [59] [57]. Furthermore, it has been suggested that follicular T cells (Tfh cells) require LAT1 for full differentiation, as indicated by impaired T cell mitogen-activated protein kinase-dependent immunoglobulin G (IgG)1 responses in *Slc7a5*^fl/fl^CD4-Cre mice [59]. Finally, there is evidence that depletion of ASCT2 results in impaired CD4^+^ memory T cell compartments [57]. T cells also express a number of alternative glutamine transporters (SLC3A2, SLC7A5 and SLC38A1) [60], but their role in T cell biology remains to be determined. 

Another substrate for LAT1, as well as SNAT2, is the essential AA methionine. Other methionine transporters expressed in T cells include SLC1A5, SLC7A8 and SLC38A1 [66,67]. Methionine metabolism is required for T cell activation and differentiation, and blocking LAT1-mediated methionine import has been shown to largely impair protein synthesis and RNA and histone methylation [68]. Of note, the tryptophan-derived metabolite kynurenine can also be transported via LAT1 [69]. Kynurenine is a potent immunomodulatory metabolite that acts as a ligand for the aryl hydrocarbon receptor (AHR) transcription factor complex, promoting CD4^+^ T cell differentiation into anti-inflammatory Tregs [70,71]. AHR itself can initiate increased uptake of kynurenine by upregulating alternative kynurenine transporters SLC7A8 and SLC36A4 [72], resulting in a positive feedback loop. Kynurenine has also been found to accumulate and activate mTORC1 in T cells in systemic lupus erythematosus (SLE), markedly enhancing T cell activity and skewing lineage development [73,74]. Conversely, SLE T cells diminish cysteine generation because of an impaired pentose phosphate pathway (PPP), resulting in decreased glutathione synthesis and accumulation of cystine. Glutathione generation can be rescued by PPP-independent treatment with the glutathione precursor N-acetylcysteine (NAC), which can also potently inhibit NADPH-dependent kynurenine synthesis, substantially replenishing intracellular reduced nicotinamide adenine dinucleotide phosphate NADPH levels. As an mTORC inhibitor [75], NAC could ultimately be used as a therapeutic in SLE [76]. NAC is transported across the plasma membrane via the glutamate transporter 1 (GLT-1/SLC1A2) [77], however not much is known about GLT-1 expression in T cells. Of note, recently it has been found that the kynurenine pathway also contributes to tumor-associated CD4^+^ T cell exhaustion [78]. 

Differently from the essential AA methionine, arginine is only conditionally essential, as it can be synthesized from citrulline [79]. In addition to synthesis inside the cell, arginine is imported by cationic amino acid transporters 1-4 (CAT1-4/SLC7A1-4) [80] and subsequently metabolized into a variety of downstream products with multiple functions. In activated T cells, imported arginine is primarily metabolized into ornithine, putrescine, agmatine and, to a lower extent, into spermidine and proline [81]. L-arginine metabolism is required for T cell survival, proliferation and anti-tumor effector function, and expression of CAT1 is upregulated upon T cell activation and if arginine levels inside the cell are low [81]. Accordingly, depletion of L-arginine results in arrest of the cell cycle in activated T cells [82], but T cells can escape arginine depletion by increasing citrulline uptake through LAT1, if available, and subsequently recycling arginine from citrulline utilizing the enzymes argininosuccinate synthase (ASS) and argininosuccinate lyase (ASL) [83]. The exact mechanisms involved in CAT1 upregulation, and its direct impact on T cell biology, remain, however, to be identified. 

The relevance of most AA transporters in T cell biology relies on their involvement in mTORC1 signaling, but other pathways, like a partly anti-inflammatory kynurenine pathway, are also emerging. As autophagy has been shown to be associated with AA transport, it might be another candidate for an AA-mediated mechanism relevant in T cell activation and differentiation [84,85]. Together, these results show that the function of AA transporters in T cells is not only to provide them with nutrients, but also to dictate T cell fate and function. 

## 3. Expression and Importance of Metabolite Transporters in Macrophages

As T cells, macrophages undergo drastic metabolic changes throughout activation, and metabolic needs of these cells depend on their polarization and function. While classically activated pro-inflammatory M1 macrophages have a broken TCA cycle and generate ATP through glycolysis and the PPP, anti-inflammatory M2 macrophages upregulate OXPHOS and FAO to meet their energetic needs [86]. As for T cells, expression of metabolite transporters has been suggested to regulate macrophage activation and polarization and is controlled by activation stimuli. However, the importance of metabolite transporters in macrophage functionality has been studied in less detail (Figure 2).

### 3.1. Glucose Transport

As glycolysis and PPP are energetically not very efficient, M1 macrophages increase their glucose consumption by upregulation of GLUT1 upon lipopolysaccharide (LPS) stimulation [87]. GLUT1 is the rate-limiting protein for glucose uptake, and its overexpression results in a hyper-inflammatory phenotype in macrophages [87]. On the other hand, deletion of GLUT1 leaves pro-inflammatory macrophages functionally impaired, while boosting M2-associated markers. GLUT1 deficiency also impaired phagocytosis and resulted in a larger abundance of anti-inflammatory metabolites ornithine and polyamines, while accumulation of pro-inflammatory metabolites, such as succinate, is reduced in macrophages lacking GLUT1 [88]. Interestingly, while downregulated in classically activated macrophages [87], expression of the glucose transporter GLUT3 is induced in macrophages in hypoxic conditions in atherosclerotic lesions, increasing lipid synthesis from the imported glucose, and possibly supporting foam cell formation [89]. Another glucose transporter, GLUT6, is expressed in macrophages. However, despite its upregulation upon LPS stimulation, it is believed that GLUT6 merely supports GLUT1-mediated glucose uptake [90]. GLUT1 thus represents the most important glucose transporter controlling macrophage activation and effector functions.

### 3.2. Lactate Transport

In innate immune cells, lactate is produced and exported in large amounts during pro-inflammatory responses [6]. Lactate export is facilitated by MCT4, whose expression is upregulated in classically activated macrophages in a myeloid differentiation primary response 88 (MYD88)- and NF-κB-dependent manner [91]. MCT4 upregulation is crucial for full macrophage activation, as its deletion results in intracellular accumulation of lactate and decreased glycolysis [91]. Macrophages also express SLC5A12, however blocking this lactate transporter does not affect macrophage metabolism or activation, suggesting SLC5A12 does not play an important role in macrophages [33]. 

Macrophages also express MCT1, MCT2 [92] and GPR81 to facilitate transport of lactate. Lactate uptake via GPR81 has been shown to reduce LPS-induced IL-1β in murine macrophages, through inhibition of NF-κB and the inflammasome [93]. Similarly, GPR81-independent lactate uptake through MCT1 inhibits pro-inflammatory signaling in bone marrow-derived macrophages and decreases macrophage activation and cytokine production [94]. Mechanistically, lactate uptake in macrophages has been shown to stabilize hypoxia-inducible factor 1α (HIF1α), and to induce the production of vascular endothelial cell growth factor (VEGF) [95]. This is especially remarkable and somehow surprising, as HIF1α is usually considered a pro-inflammatory mediator [96]. Importantly, studies with primary human monocytes and peripheral blood mononuclear cells have confirmed these observations to be relevant for human macrophages biology as well [97]. Overall, lactate import in macrophages seems to limit their pro-inflammatory potential.

### 3.3. Fatty Acid Transport

The FA transporter FATP1 is downregulated in classically activated macrophages, and its deletion results in increased glucose metabolism and a pro-inflammatory phenotype in adipose tissue macrophages [98]. In accordance, FA uptake through FATP1 is crucial for alternative macrophage activation, and overexpression of FATP1 limited macrophage-induced inflammation through a reduction in glycolysis and PPP [98]. Importantly, *Fatp1* expression in macrophages is largely controlled by peroxisome proliferator-activated receptor (PPAR)α and PPARγ [99]. In contrast, LPS-stimulated macrophages increase expression of the medium-chain FA transporter GPR84, initiating FA uptake and increasing IL-12 p40 expression [46]. More recently, GPR84-mediated FA uptake has been found to induce an inflammatory CD11b^hi^ status in alveolar macrophages during lung injury [100], highlighting the complex functions of FA in macrophage activity.

### 3.4. Amino Acid Transport

Like T cells, classically activated macrophages have an increased demand for AAs to synthesize proteins and feed into metabolic pathways, however transport of AAs in macrophages has been less studied than in T cells. Classically activated macrophages increase expression of LAT1 to increase leucine uptake, contributing to mTORC1-induced metabolic reprogramming and increased cytokine production. Blockage or downregulation of LAT1 in macrophages results in decreased mTORC1-induced IL-1β production [101]. Expression of arginine transporter CAT2 is induced in both classically and alternatively activated macrophages, and L-arginine is subsequently imported. Deletion of CAT2 notably reduced arginine metabolism, resulting in decreased amount of nitrites, polyamines and proline, while arginase and nitric-oxide synthase 2 (NOS2) activity remained unchanged [102]. This directly contradicts a previous study, claiming NOS2 activity to depend on CAT2 expression [103].

Mouse peritoneal macrophages upregulate expression of cystine/glutamate transporter (XCT/SLC7A11) upon LPS stimulation [104] to import cystine, which is subsequently converted to cysteine for glutathione and protein synthesis [105]. The same system, as well as the X_AG_ system, was found to import glutamate in human monocyte-derived macrophages for production of glutathione [106]. Glutathione is particularly important in macrophages for maintaining the thiol redox state and protecting from oxidative stress [107]. 

Human macrophages also import L-carnitine via organic cation transporter, novel and type 1 and 2 (OCTN1/SLC22A4 and OCTN2/SLC22A5). L-carnitine is known to mediate differentiation of monocytes into macrophages [108], and it has been suggested that deficiencies in carnitine transport are associated with increased pathogenicity in Crohn’s disease [109]. 

## 4. Expression and Importance of Metabolite Transporters in Other Immune Cells

T cells and macrophages are by far the most investigated immune cells in the field of immunometabolism. However, the role of metabolic reprogramming in the fate and function of other cells of the immune system cannot be denied. Nonetheless, very little is known about the role of metabolite transporters in these cells, but it can generally be assumed that expression of glucose importers correlates with cellular activation. Accordingly, B cells upregulate GLUT1, 3 and 4 when activated [18], and restriction of glucose uptake results in attenuated activation and impaired antibody production [110]. Upregulation of GLUTs is also critical for functional activation of dendritic cells (DCs) [111], neutrophils [112] and NK cells [113].

Regarding FAs and AAs, butyrate and propionate uptake from commensal gut bacteria via SLC5A8 has been shown to induce a tolerogenic phenotype in DCs in the gut, protecting against intestinal inflammation [114]. Moreover, some inflammatory neutrophils in cystic fibrosis airways have been found to upregulate ASCT2 to facilitate AA uptake [112], confirming an important role for metabolite transporters in other immune cell subsets. Although metabolite uptake undoubtedly occurs in these cells, information about the expression and function of metabolite transporters in immune cells other than T cells and macrophages is still limited.

## 5. Metabolite Transport as Intercellular Communication

Metabolites function as signaling molecules both inside and outside cells. While many metabolite-induced signaling events are receptor-mediated, active metabolite transport is used by certain cell types to change their environment and manipulate the behavior of surrounding cells. In addition to immune regulation in T cells and macrophages, non-immune cells use active metabolite transport to shape immune responses by exporting or scavenging metabolites. By creating a favorable milieu, some cells can use metabolites to signal to immune cells and influence their activity. 

Prostaglandin E2 (PGE2) can be translocated by SLCO2A1, a transporter expressed in murine macrophages. SLCO2A1 exports PGE2 from the cell in inflammatory conditions [115], potentially to maintain removal of neutrophils from inflamed tissue and promote inflammation resolution [116]. How PGE2 export is initiated and what mechanisms are involved remain to be investigated.

Short-chain FAs (SCFAs) are produced by commensal microbiota in the gut and contribute to intestinal homeostasis. T cells take up SCFAs produced by the microbiome via GPR43 and promote microbiota antigen-specific IL-10 production, ultimately suppressing immune responses against commensal bacteria [44]. Furthermore, microbial butyrate, an SCFA, has been found to have a similar effect on macrophages in the gut by inducing a homeostatic, M2-like phenotype [117].

Classically activated macrophages produce and release succinate, a pro-inflammatory metabolite that induces IL-1β production through stabilization of HIF-1α [118] and reactive oxygen species (ROS) production [119]. Succinate binds the succinate receptor SUCNR1, and in neural stem cells (NSCs), binding of SUCNR1 triggers an anti-inflammatory program, characterized by release of the immune modulator PGE2, and expression of the succinate transporters SLC13A3 and SLC13A5. These transporters subsequently take up the extracellular succinate, effectively scavenging it from the extracellular space and thereby reducing succinate-mediated inflammation. Through this mechanism, NSCs are able to actively reduce neuroinflammation in the central nervous system, and might protect from chronic inflammation and degenerative neurological diseases [120].

Another example for metabolite-induced reprogramming of immune cells is the reversed Warburg effect. The Warburg effect describes a metabolic state in cancer cells that is characterized by aerobic glycolysis and results in production of lactate and pyruvate [4]. In the reversed Warburg effect, cancer cells induce the Warburg effect in neighboring fibroblasts. These “corrupted” cells then produce large amounts of lactate and pyruvate and export them to the extracellular space through mono-carboxylate transporters MCT1 and MCT4 [121]. Cancer cells subsequently express MCT1 and MCT4 to take up these energy-rich metabolites and feed them into their TCA cycle to generate ATP [122]. At the same time, the exported lactate also diminishes CTL responses in the tumor microenvironment due to the inhibitory effects of lactate on glycolysis and subsequently CTL activation [29], and initiates the generation of tumor-tolerating, anti-inflammatory Tregs [32]. This is mediated by reduced T-box expressed in T cells (T-bet) and increased forkhead box protein P3 FoxP3 expression in tumor-associated CD4^+^ T cells [123]. A similar effect can be observed in macrophages, where lactate produced by cancer cells and fibroblasts induces expression of VEGF in tumor-associated macrophages, polarizing them into a homeostatic, M2-like phenotype in a HIF1α-dependent manner [95]. A great example for lactate as a communicator among cells in a non-cancerous context was recently described by Zhang and colleagues: in the ischemic muscle, endothelial cells export lactate through MCT4. Macrophages then import the lactate via MCT1, subsequently acquiring an M2-like, anti-inflammatory phenotype, promoting muscle regeneration [124].

Interestingly, extracellular lactate can also induce autoimmunity. In RA, synovial fibroblasts upregulate MCT4, much like in the tumor environment, and export large amounts of lactate, enriching the synovial fluid and acidifying it [125]. As discussed before, lactate import into CD4^+^ T cells can induce production of IL-17. In RA, this increased cytokine production widely contributes to the pathology in this disease. Furthermore, as T cell motility is impaired by lactate as well, these T cells are retained at the site, further promoting inflammation [33]. These works thus highlighted the potential of extracellular metabolites to act as signaling molecules, affecting both tissue and immune cell functionality.

## 6. Metabolite Transport as Therapeutic Target

As described above, regulation of metabolite transport in immune cells is highly correlated with cell fate and activity, and many pathologies can be linked to transporter malfunctions. Furthermore, many new diseases, mostly of monogenetic origin, keep being discovered and linked to transporter malfunctions or mutations, and the demand for drugs impacting such pathologies is high. Notably, approximately 20% of known SLC transporters are associated with human disease through mutation [7]. As the scientific and clinical relevance of metabolite transporters is increasing, new therapeutic strategies are under development that specifically target these essential proteins.

It has been shown that monocytes from RA patients overexpress LAT1, alongside increased cytokine production, suggesting increased AA influx to be linked to chronic inflammation [101]. Recently, blocking LAT1 expressed on T cells has been found to successfully reduce inflammation in psoriasis [126]. Drugs targeting LAT1 are currently undergoing clinical trials as cancer therapeutics [127]. If successful, these LAT1 inhibitors could potentially be used to treat autoinflammatory diseases. As mentioned before, tumor-associated Tregs upregulate expression of CD36, an FA translocase. CD36 has been found to be crucial for lipid metabolism in these cells and enables them to suppress immune responses in the tumor microenvironment [41]. Targeting CD36 with antibodies successfully depleted pro-tumor Tregs and improved tumor clearance without inducing autoimmunity [41]. CD36 might therefore be an interesting target for new cancer immunotherapies. Another FA transporter, GPR84, is currently being investigated as a potential therapeutic target in fibrotic diseases. The GPR84-blocking compound PBI-4050 has been shown to not only decrease FA-induced activation of fibroblasts to profibrotic myofibroblasts, but also to reduce macrophage activation [128]. Notably, PBI-4050 has recently undergone phase 2 clinical trials for the treatment of idiopathic pulmonary fibrosis [129]. Finally, as previously described, blocking of lactate transporter SLC5A12 may prove successful in the treatment of RA by limiting T_H_17-induced inflammation. Pucino and colleagues have committed to developing potent monoclonal antibodies and advancing to pre-clinical studies [33].

Another potential target for metabolite transporter-modulating drugs are cancer cells. Inhibiting transporters of glucose, AAs or lactate may indeed represent a powerful tool to decrease tumor growth, as these metabolites are usually in high demand in cancer cells, and their respective transporters are often overexpressed [130,131,132]. Inhibitors of MCT1 in tumor cells have been shown to effectively inhibit lipid synthesis in cancer cells and increase influx of tumor immune cells, and some of these inhibitors are currently being tested in clinical trials [133]. On the other hand, block or downregulation of the lactate exporter MCT4 could potentially decrease lactate build-up in the tumor microenvironment, thus reducing immunosuppression. Studies in multiple types of cancers have shown that this approach not only inhibits tumor growth [134] and prolongs relapse time [135], but also induces apoptotic programs in the cancer cells themselves [136]. Another interesting example is the anti-inflammatory RA drug Sulfasalazine. Sulfasalazine blocks the cystine–glutamate exchange transporter XCT that has also been found to be overexpressed in tumor cells [137]. Sulfasalazine is currently being tested in various tumor models, showing promising results in enhancing tumor clearance in prostate cancer [138,139]. Combined blockage of metabolite transporters in tumor cells may thus offer a new strategy for cancer therapy. Overall, these findings highlight the potential of metabolite transporters as novel drug targets for immunomodulation and cancer therapy. 

## 7. Conclusions

Metabolic pathways shape every aspect of immune cell fate and function. This is becoming more evident with every new publication in the field of immunometabolism. Metabolite transporters, as metabolites themselves, have acquired novel functions as regulators of immunity, some of which we have presented in this review.

As evident form the literature presented, active metabolite transport is key in organizing immune cell functionality. Regulation of metabolite transporter expression is also an important tool in intercellular communication, a system that is susceptible to manipulation and can thus easily promote disease or cancer. Conversely, targeting metabolite transporters bears a large potential for drug development, and has been shown to reduce pathologies in many diseases. While functions of the transporters highlighted in this review are well documented, many metabolite transporters are still categorized as orphan transporters without known function or substrates. Due to a lack of crystallographic data for many SLCs, specificity of many potential and known transporters is hard to determine [140]. On the other hand, many immunomodulatory metabolites, especially from the TCA cycle [141], are known to be actively transported among cells with transporters yet to be identified. It is highly plausible that future research will reveal many more SLCs to be involved in metabolite transport, further diversifying their relevance as regulators of immunity. 

## Figures and Tables

**Figure 1 metabolites-10-00418-f001:**
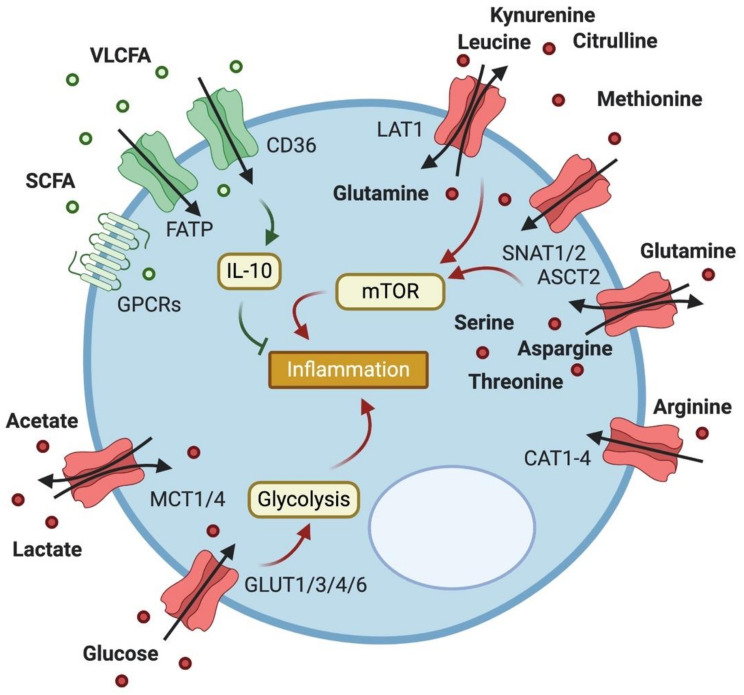
Expression of metabolite transporters on T cells and their impact on T cell function. T cells express a variety of transporters to facilitate their metabolic needs. Import of fatty acids (FAs) of various sizes reduces inflammation through FA oxidation and IL-10 production, while import of lactate and acetate boosts inflammation. Transporters of the GLUT family take up glucose to increase glycolysis in activated T cells, while amino acid (AA) transport drives inflammation mainly via mTOR activation.

**Figure 2 metabolites-10-00418-f002:**
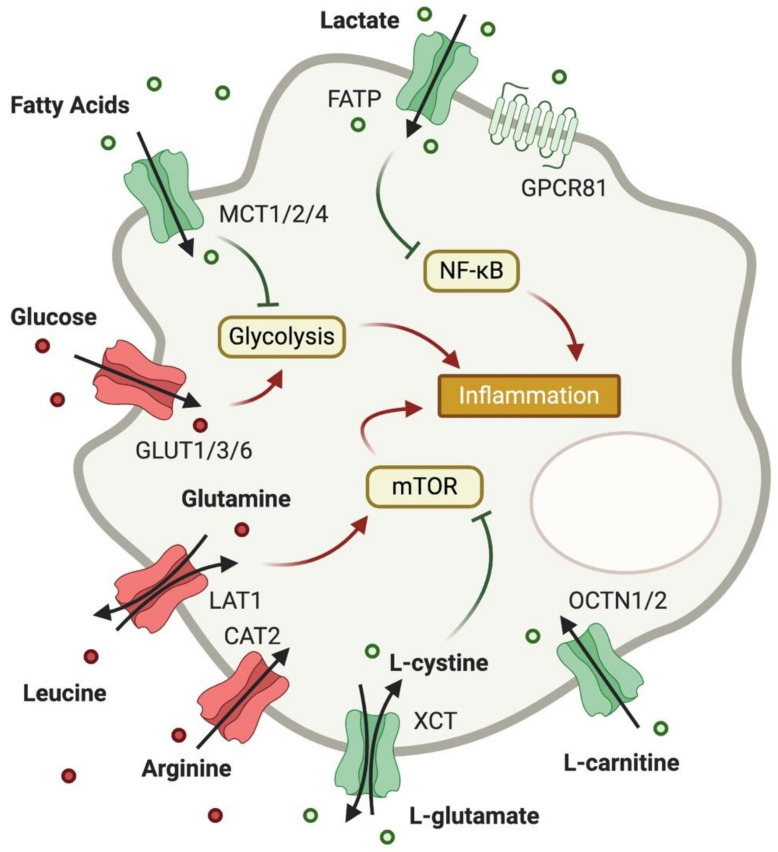
Expression of metabolite transporters on macrophages and their impact on macrophage function. As in T cells, glucose uptake is increased in pro-inflammatory macrophages, and AA uptake boosts mTOR activity. Cystine and glutamate, however, can limit mTOR activation through generation of cysteine and glutathione. Carnitine also exhibits anti-inflammatory potential, while FA uptake can limit glycolysis. Unlike in T cells, lactate uptake into macrophages is considered to be anti-inflammatory through inhibition of NF-κB.

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
