# Peer review of "Metabolite Transporters as Regulators of Immunity"

_metabolites, 2020, doi:10.3390/metabo10100418_

Round 1

Reviewer 1 Report

From the perspective of this reviewer, more clarity is needed about the role of certain metabolites, such as kynurenine, cysteine, glutathione and their connection to immunity and autoimmunity.

Although its relationship to activation of mTORC1 is mentioned, the very paper that described that relationship has not been cited or discussed. Kynurenine has been found most accumulated metabolite in patients with SLE, the prototypical systemic autoimmune disease in humans, and it was found to activate mTORC1 (Metabolomics. 11:1157-1174; https://pubmed.ncbi.nlm.nih.gov/26366134/). Blockade of mTORC1 with rapamycin (Lancet, 391:1186-1196; https://pubmed.ncbi.nlm.nih.gov/29551338/) or NAC is therapeutic in SLE (Arthritis Rheum. 64: 2937-2946; https://pubmed.ncbi.nlm.nih.gov/22549432/). The importance of the kynurenine pathway has been confirmed in patients (https://doi.org/10.1371/journal.pone.0159384) and mice with SLE (https://stm.sciencemag.org/content/12/551/eaax2220.abstract).

Since NAC is replacing diminished cysteine in lupus T cells, it would be important to discuss the relationship of kynurenine, cysteine, and glutathione to mTOR pathway activation both in terms of their membrane transport and hierarchy of metabolic fluxes. These pathways and metabolite transporters should be depicted in mechanistic diagrams that would markedly enhance the impact of this paper.

Reviewer 2 Report

I would like to congratulate the authors to a well written, comprehensive and fairly complete review of relevant literature on metabolite transporters and their role in the immune system. 

Minor comments:

Line 370: Wang et al – should be reference style as Arabic number in brackets. The citation order may also be out of place here as a result.

Same for line 393 – should be a reference in brackets. Gout et al is not listed in the Bibliography.

“Acknowledgments: TBC” this should be added.

I suggest to summarise the descriptions of various effects of groups of metabolite transporters in a graphic figure, which could include cell type specific effects, or visualize key principles of transporters for Immunity.

Round 2

Reviewer 1 Report

None.